# MonkeySee: Space-time-resolved reconstructions of natural images from macaque multi-unit activity

**Lynn Le[1], Paolo Papale [2], Katja Seeliger[3], Antonio Lozano[2], Thirza Dado[1],
Feng Wang [2], Pieter Roelfsema [2,4,5,6], Marcel van Gerven[1], Yağmur Güçlütürk[1], Umut Güçlü†[1]**

[1] Donders Institute for Brain, Cognition and Behaviour, Radboud University, Nijmegen, Netherlands
[2] Netherlands Institute for Neuroscience, Amsterdam, Netherlands
[3] Max Planck Institute for Human Cognitive and Brain Sciences, Leipzig, Germany
[4] Centre for Neurogenomics and Cognitive Research, Vrije Universiteit, Amsterdam, Netherlands
[5] Institut de la Vision, Paris, France
[6] Amsterdam University Medical Center, Amsterdam, Netherlands

†Correspondence Email: u.guclu@donders.ru.nl

## Abstract

In this paper, we reconstruct naturalistic images directly from macaque brain signals using a convolutional neural network (CNN) based decoder. We investigate the ability of this CNN-based decoding technique to differentiate among neuronal populations from areas V1, V4, and IT, revealing distinct readout characteristics for each. This research marks a progression from low-level to high-level brain signals, thereby enriching the existing framework for utilizing CNN-based decoders to decode brain activity. Our results demonstrate high-precision reconstructions of naturalistic images, highlighting the efficiency of CNN-based decoders in advancing our knowledge of how the brain's representations translate into pixels. Additionally, we present a novel space-time-resolved decoding technique, demonstrating how temporal resolution in decoding can advance our understanding of neural representations. Moreover, we introduce a learned receptive field layer that sheds light on the CNN-based model's data processing during training, enhancing understanding of its structure and interpretive capacity.

## 1 Introduction

Artificial neural network models designed for decoding naturalistic images from neural activity signals significantly advance our understanding of how visual information is processed in the brain. Decoding models aim to disentangle patterns of neural responses to different stimuli, offering insights into how visual stimuli (e.g., a brown horse or a white t-shirt) are represented by neural populations. Leveraging a large amount of naturalistic data facilitates the reconstruction of natural vision and allows for comprehensive analyses of visual features. Due to the immense variety of naturalistic visual space, reconstructing such stimuli from neural activity is considered the most challenging but also the most intriguing problem in neural decoding.

Convolutional neural networks (CNNs) have recently become a cornerstone in neural decoding and encoding models. Their ability to study fundamental features carried by populations of neuronal signals has led to significant advancements in understanding the computational mechanisms of natural scene perception. Encoding models provide valuable insights into how brain activity changes in response to stimuli, showcasing the features of CNNs that predict neural responses within specific

38th Conference on Neural Information Processing Systems (NeurIPS 2024).

brain regions. Conversely, decoding models reveal the information content within brain areas without making assumptions about the representation of that information.

Despite the prowess of CNNs in distinguishing between different brain signal patterns, their effectiveness in decoding shifts in neuronal activity remains limited. This limitation is crucial – a lack of variance in reconstructed images with changing brain signals suggests a disconnect between the models and the brain's interpretive functions. This performance gap could stem from intrinsic model limitations or data distribution issues. Refining these methods is essential to enhance our understanding of how closely decoding models can approximate the complex processes underlying visual perception.

While CNNs are engineered to handle complex, multi-dimensional data, including spatial (height and width) and depth (color) dimensions, there remain significant distinctions between CNN processing and human brain information processing. To address these challenges, we have developed a fully convolutional decoding model trained from scratch using the THINGS dataset – a highly diverse collection of naturalistic stimuli. This dataset enriches the model's exposure to varied features, which is crucial for interpreting brain representations accurately.

In this paper, we present the following contributions:

- **Homeomorphic decoder**: We propose a CNN-based decoder trained to investigate the importance of spatial and temporal information carried in neuronal signals for high-fidelity visual reconstructions.

- **End-to-end inverse retinotopic mapping**: We integrate an interpretable layer in the model, known as the end-to-end inverse retinotopic mapping. This layer dynamically learns to map brain signals to a 2D image during training. The adaptive mechanism of this layer, influenced by the entire learning process, allows the decoder to organize its own input spatially.

- **Model inference and analysis**: By performing model inferences with truncated brain data, our approach dissects how the network reorganizes its weights based on spatially separated brain regions for reconstructions. This method aligns with known neuroscientific principles, enhancing the interpretability of decoded features.

- **Temporal dynamics**: Our model incorporates specific time intervals for neuronal signal input, aligned with latency periods observed in the ventral visual pathway (V1 to IT). This temporal aspect allows for a deeper analysis of how visual processing evolves over time.

The remainder of this paper is structured as follows. Section 2 reviews related work in neural decoding using CNNs, highlighting advances and existing challenges. Section 3 details the materials and methods, including data acquisition, preprocessing, and the model architecture. Section 4 presents the results and discussion, evaluating the performance and interpretability of our model. Finally, Section 5 concludes the paper and suggests avenues for future research.

## 2  Related work

Deep neural networks (DNNs) and generative adversarial networks (GANs) have recently achieved notable success in decoding visual information from brain activity, particularly using fMRI data [1, 2, 3, 4, 5, 6, 7, 8]. Seeliger et al. (2018) [5] utilized GANs to reconstruct grayscale images and handwritten characters from fMRI data, demonstrating the versatility of adversarial methods. Nishimoto et al. (2011) [1] used voxel-wise modeling to reconstruct complex video stimuli from brain activity, focusing on capturing detailed neural responses in the visual cortex. Shen et al. (2019a) [8] further advanced this by training end-to-end models for image reconstruction, incorporating high-level feature losses into the GAN framework.

Reconstructing dynamic stimuli like videos presents additional challenges. Han et al. (2019) [6] used variational auto-encoders for video reconstruction, achieving low-level property reconstructions but struggling with finer details. This highlights the complexity of decoding dynamic visual information.

CNNs are effective for neural decoding due to their capability to process complex, multi-dimensional data. Sarraf and Tofighi (2016) [9] treated fMRI slices as separate images, but this method was limited by noise and did not respect neural topography. Approaches using 3D convolutions to preserve spatial

structure [10], and geometric deep learning on cortical meshes [11, 12, 13, 14, 15], have shown promise in capturing brain features more effectively.

Decoding naturalistic stimuli using large datasets pushes conventional model boundaries [16, 17]. Le et al. (2022) [17] reconstructed images and videos from fMRI data by converting voxel responses into 2D representations aligned with the visual field, then applying fully convolutional networks trained with VGG feature loss and adversarial regularizers. This method showed significant improvements over previous techniques.

Our work uses multi-unit activity (MUA) data from Utah electrode arrays in the macaque visual cortex (V1, V4, IT), which offers higher temporal resolution and captures fine-grained neuronal activity compared to fMRI. Bashivan et al. (2019) [18] integrated receptive field concepts into neural network models for better interpretability. Building on this, we incorporate an end-to-end inverse retinotopic mapping layer within our convolutional decoder. This layer dynamically maps brain signals to 2D images, improving spatial feature organization and providing deeper insights into neural processing.

## 3 Material & methods

### 3.1 Data

We used images from the THINGS database [19], containing high-resolution images across various object categories. Each image had three color channels (RGB) and was resized to $96 \times 96$ pixels to meet model input requirements and reduce computational complexity. Images were presented in the lower right quadrant, shifted 150 pixels right and down from the central fixation point.

A passive fixation task was conducted with a 7-year-old male macaque (*Macaca Mulatta*) across 22,348 trials (22,248 training and 100 test trials). The macaque maintained fixation on a central dot while images, presented in a randomized sequence of four per trial, were displayed for 200 ms with a 200 ms gray screen interval. The macaque was rewarded with juice for maintaining fixation. Ethical approval was obtained from the Royal Netherlands Academy of Arts and Sciences, adhering to the NIH Guide for the Care and Use of Laboratory Animals, ensuring the macaque's well-being and minimizing stress.

MUA was recorded from 15 Utah electrode arrays implanted in V1 (7 arrays), V4 (4 arrays), and IT (4 arrays), capturing neural activity at 1 ms resolution over 200 ms per trial. Electrodes were selected using a self-correlation reliability score with a threshold of 0.4, reducing the original 1024 electrodes to 576 (including losses from a broken electrode). Neural responses were z-scored (mean subtracted, divided by standard deviation). Time windows were defined as 0–125 ms for V1, 25–150 ms for V4, and 50–175 ms for IT. A 25 ms smoothing window was applied, and data were temporally downsampled to either 8 Hz (averaged over 125 ms) or 40 Hz, ensuring consistent and normalized data for modeling.

### 3.2 Models

In this section, we describe the models used for decoding the neural responses into visual stimuli. The main model is a homeomorphic decoder, and we compare its performance against a baseline decoder. Additionally, we employ a discriminator to facilitate adversarial training.

#### 3.2.1 Homeomorphic decoder

The homeomorphic decoder transforms neural responses into retinal embeddings and subsequently reconstructs the visual stimuli from these embeddings. The architecture leverages several neural network components, including pre-trained and end-to-end trained models.

**Pre-trained inverse retinotopic mapping** The first variant of our homeomorphic decoder uses a pre-trained CNN to perform inverse retinotopic mapping. The model projects neural responses onto retinal embeddings using learned weights and subsequently reconstructs the visual image from these embeddings. The process involves two types of embeddings: *spatial embeddings* and *spatiotemporal embeddings*.

For *spatial embeddings*, the retinal embedding $E$ is computed directly from the neural responses at a single timepoint:

$$E_{axy} = \sum_{e \in \text{MEA}_a} r_e W_{exy}$$

where $r$ represents the neural responses over $576$ electrodes at a single timepoint, and $W_{exy}$ are the learned spatial weights for mapping the neural responses to retinal embeddings for each microelectrode array ($\text{MEA}_a$).

For *spatiotemporal embeddings*, the retinal embedding $E$ is computed from the neural responses over multiple timepoints:

$$E_{i_t xy} = \sum_{e \in \text{ROI}_i} R_{et} W_{exy}$$

where $R$ represents the neural responses over $576$ electrodes across 5 timepoints, and $W_{exy}$ are the learned weights for mapping the neural responses to retinal embeddings for each region of interest ($\text{ROI}_i$).

The weights $W$ are optimized by minimizing the following objective function:

$$W_e^*, \alpha_e^* = \min_{W, \alpha} \|r_e - \hat{r}_e\|_2 + \lambda_1 \|W\|_1 + \lambda_2 (\|W\|_2^2 + \|\alpha\|_2^2) + \lambda_3 \Delta \alpha$$

where $\hat{r}_e = \sum_c \alpha_{ec} \sum_{xy} W_{exy} f(S)_{cxy}$ is the estimated neural response with $f$ representing activations from a pre-trained Inception v1 network and $\Delta w$ is the Laplace operator. Here, $W$ can be considered the spatial weights of interest and $\alpha$ the feature weights. The specific layers of the network used for different cortical areas are: `conv2d2` for V1, `mixed4a` for V4, and `mixed4d` for IT. These embeddings leverage the pre-trained network to efficiently map neural activity patterns to their corresponding visual stimuli representations, forming a crucial component of our decoder.

**End-to-end trained inverse retinotopic mapping**   This variant of our homeomorphic decoder involves end-to-end training of the inverse retinotopic mapping from neural responses $R$ to retinal embeddings $E_a$, allowing the model to learn optimal parameters directly from data without pre-trained CNN features.

The inverse receptive field (see Figure 6) computes $E$ as:

$$E_{axy} = \sum_{e \in \text{MEA}_a} r_e \exp\left(-\left(\frac{(x - x_e)^2}{2\sigma_e^2} + \frac{(y - y_e)^2}{2\sigma_e^2}\right)\right)$$

where $r_e$ represents the neural responses, $x_e$ and $y_e$ denote the spatial coordinates of electrode $e$, and $\sigma_e$ represents the standard deviation parameter that determines the spatial spread of the receptive field associated with each electrode. The parameters $x_e$, $y_e$, and $\sigma_e$ are learned during training.

Once $R_a$ is mapped onto $E_a$, a pixel-to-pixel U-Net reconstructs the stimulus $S_{\text{fake}} = \text{U\_Net}(E_a)$, where $S_{\text{real}}$ is the ground-truth stimulus. Loss components are functions of $S_{\text{real}}$ and $S_{\text{fake}}$, incorporating $E_a$ into the objective function.

This end-to-end approach adapts to the specific characteristics of neural responses and visual stimuli, improving reconstruction performance.

**Pixel-to-pixel mapping**   The final component of our homeomorphic decoder employs a U-Net architecture designed for pixel-to-pixel mapping of retinal embeddings to visual stimuli. The U-Net model is a powerful neural network architecture commonly used for image segmentation tasks due to its ability to capture both local and global image features through its contracting and expansive paths connected via skip connections. We are using a standard U-Net architecture; for details refer to Appendix A.1.

### 3.2.2   Baseline decoder

We employ a baseline decoder as a reference for performance evaluation. This simpler model transforms neural responses directly into visual stimuli, without the use of intermediate retinal embeddings. It adopts a modified U-Net architecture, retaining only the expansive path from the homeomorphic decoder while omitting the contracting path and skip connections. The input is a

$576 \times 1$ tensor representing neural responses from 576 electrodes at a single timepoint, and the output is a RGB $\times 96 \times 96$ visual stimulus. Despite its simplicity, the baseline serves as a crucial benchmark for assessing more complex models like the homeomorphic decoder. This approach is inspired by [8], which is regarded as a state-of-the-art reconstruction model [20].

### 3.2.3 Discriminator

The discriminator plays an integral role in the training process by differentiating between real and fake visual stimuli, thus facilitating adversarial training. We employ a modified U-Net architecture, using only the contracting path, to serve as our discriminator. The input to the discriminator is the visual stimulus, $S$, of dimensions RGB $\times 96 \times 96$. The output is a scalar probability $p$, indicating the likelihood that the given image is real. Adversarial training, where the discriminator aims to distinguish real images from reconstructed images, drives the decoder to generate more realistic and accurate visual stimuli. The loss calculated from the discriminator's assessments is crucial for improving the fidelity of the decoded images.

## 3.3 Training

This section outlines the optimization procedures, loss functions, and strategies used to train the decoders and the discriminator for high-fidelity reconstruction of visual stimuli from neural responses. Source code is available on our GitHub repository[1].

### 3.3.1 Training parameters

The dataset comprised 22,348 training samples and 100 test samples, which were exclusively used for testing and never during training. We used the Adam optimizer with a learning rate of 0.002 and beta coefficients of 0.5 and 0.999 to ensure convergence. The loss function included discriminator loss ($\alpha_{\text{discr}}$) at 0.01, VGG feature loss ($\beta_{\text{vgg}}$) at 0.9, and L1 pixel-wise loss ($\beta_{\text{pix}}$) at 0.09 to balance sensitivity. Training spanned 50 epochs on a Quadro RTX 6000 GPU, utilizing approximately 10,000 MiB of GPU memory.

### 3.3.2 Decoder training

The training of the decoders (both homeomorphic and baseline) involves a combination of losses designed to ensure realism and accuracy in the reconstructed images. The *adversarial loss* encourages the decoder to generate realistic images that the discriminator cannot distinguish from real images. This loss is defined as the binary cross-entropy loss between the discriminator's output and the true labels (1 for real images and 0 for generated images):

$$\mathcal{L}_{\text{adv}} = -\mathbb{E}_{S_{\text{real}}}[\log D(S_{\text{real}})] - \mathbb{E}_{S_{\text{fake}}}[\log(1 - D(S_{\text{fake}}))]$$

To further enhance the quality of the reconstructed images, we use a *feature matching loss* based on the activations of a pre-trained VGG-19 network. The feature loss is the mean squared error (MSE) between the feature representations of the real and generated images at various layers (`conv1_2`, `conv2_2`, `conv3_4`, `conv4_4`, and `conv5_4`) of the VGG-19 network:

$$\mathcal{L}_{\text{feat}} = \sum_l \|\phi_l(S_{\text{real}}) - \phi_l(S_{\text{fake}})\|_2^2$$

where $\phi_l$ denotes the feature map at layer $l$ of the VGG-19 network. The *pixel-wise loss* ensures that the reconstructions are close to the original images in pixel space. We use the mean absolute error (MAE) to quantify this loss:

$$\mathcal{L}_{\text{pixel}} = \|S_{\text{real}} - S_{\text{fake}}\|_1$$

The total loss for the decoder is a weighted sum of the adversarial loss, feature loss, and pixel loss:

$$\mathcal{L}_{\text{decoder}} = \lambda_{\text{adv}}\mathcal{L}_{\text{adv}} + \lambda_{\text{feat}}\mathcal{L}_{\text{feat}} + \lambda_{\text{pixel}}\mathcal{L}_{\text{pixel}}$$

where $\lambda_{\text{adv}}$, $\lambda_{\text{feat}}$, and $\lambda_{\text{pixel}}$ are the weights for each respective loss component.

---

[1] https://github.com/neuralcodinglab/MonkeySee

### 3.3.3 Discriminator training

The discriminator is trained to distinguish between real and generated images, using the adversarial loss defined as the binary cross-entropy loss:

$$\mathcal{L}_{\text{discriminator}} = -\mathbb{E}_{S_{\text{real}}}[\log D(S_{\text{real}})] - \mathbb{E}_{S_{\text{fake}}}[\log(1 - D(S_{\text{fake}}))]$$

### 3.3.4 Optimization strategy

The decoder and discriminator are optimized using the Adam optimizer with default parameters (learning rate = 0.0002, $\beta_1 = 0.5$, $\beta_2 = 0.999$). Early stopping is applied based on validation set performance to prevent overfitting. An image buffer maintains a history of generated images to stabilize adversarial training by varying the discriminator's inputs. During training, the decoder and discriminator are updated alternately: 1) the discriminator using adversarial loss, and 2) the decoder using the total loss (adversarial, feature, and pixel losses). Training continues until convergence, monitored by evaluation metrics.

## 3.4 Evaluation metrics

To robustly evaluate the performance of our decoding models, we employ a variety of metrics that assess the accuracy and quality of the reconstructed visual stimuli from both a feature-wise and perceptual quality perspective.

**Feature correlation** One of the primary metrics used to evaluate stimulus reconstruction is the feature correlation between the reconstructed images and the original images. We use a pre-trained AlexNet, extracting feature representations at various layers (`conv1`, `conv2`, `conv3`, `conv4`, `conv5`, `fc6`, `fc7`, `fc8`). The Pearson correlation coefficient is calculated between corresponding feature maps of the original and reconstructed images: $\rho_{\phi_l} = \text{Pearson}(\phi_l(S_{\text{real}}), \phi_l(S_{\text{fake}}))$ where $\phi_l$ denotes the feature map at layer $l$ of AlexNet. High correlation values indicate that the reconstructed images capture similar feature representations as the original images. This standard metric for evaluating reconstruction quality was also used by Le et al. (2022) [17].

**Image colorfulness** To evaluate the perceptual quality of the reconstructed images, we use the Hasler and Süsstrunk colorfulness metric [21]. This metric quantifies the colorfulness of an image, which is an important aspect of human visual perception. The metric is computed as: $C = \sigma_{rg} + 0.3\mu_{rg}$ where $\sigma_{rg}$ and $\mu_{rg}$ are the standard deviation and mean of the color difference vector $rg = (R - G)$ across the image.

**Occlusion analysis** We performed spatial and spatiotemporal occlusion analyses to assess the contributions of different spatial and temporal regions to reconstruction accuracy. In spatial occlusion, parts of the input are systematically removed to identify which regions are most critical for decoding. In spatiotemporal occlusion, specific segments of neural responses are occluded to evaluate the importance of different timepoints and spatial regions in the reconstruction process.

## 4 Results and discussion

### 4.1 Stimulus reconstruction

The performance of different decoding models in reconstructing visual stimuli from neural responses is evaluated across multiple dimensions, including model comparison, spatial occlusion analysis, and spatiotemporal occlusion analysis.

### 4.1.1 Model comparison

We compared the reconstruction performance of three variations of our homeomorphic decoder – spatial, spatiotemporal, and end-to-end inverse retinotopic mapping – against the baseline model, both qualitatively and quantitatively.

Figure 1 presents qualitative results. The spatial, spatiotemporal, and end-to-end decoders consistently outperformed the baseline, better preserving textures, shapes, and colors.

| Stimuli | Spatial | Spatiotemporal | End-to-end | Baseline |

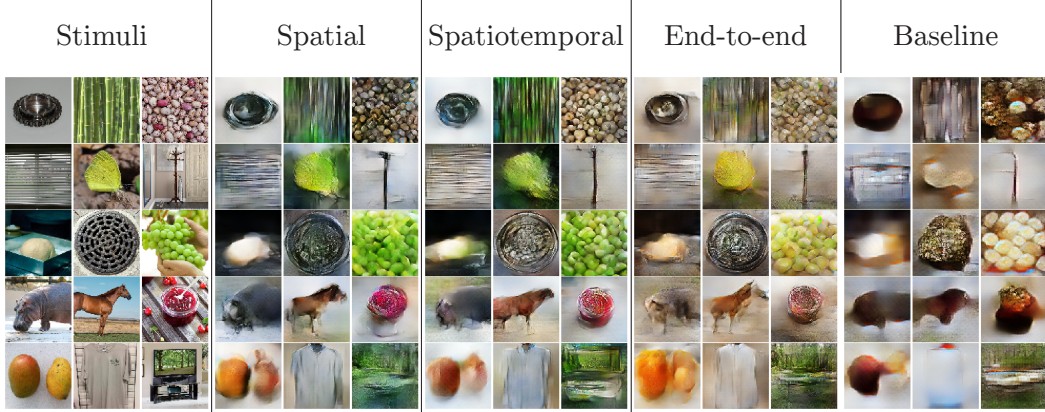

Figure 1: Sample stimuli and corresponding reconstructions from models. The "Spatial" and "Spatiotemporal" column show results from the pre-trained inverse retinotopic mapping model, explained in Section 3.2.1. The "End-to-end" column shows reconstructions from the space-resolved model with a component that learns the neuron's receptive field explained in Section 3.2.1. "Baseline" shows the reconstructions of a model we implemented explained in Section 3.2.2.

Table 1 provides a quantitative comparison using Pearson correlations between reconstructed and original images across different AlexNet layers (Section 3.4). The spatiotemporal decoding model achieved the highest feature correlation across most layers, highlighting its superior ability to capture and reconstruct the features present in the original stimuli. Specifically, the spatiotemporal model excelled in the deeper layers (`fc7` and `fc8`), which capture high-level feature representations, demonstrating the model's capability to capture both fine-grained and abstract features encoded in the neural responses.

Table 1: Feature correlations of reconstructions with original images across AlexNet layers.

|       | Spatial | Spatiotempral | End-to-end | Baseline |
|-------|---------|---------------|------------|----------|
| conv1 | 0.358   | **0.372**     | 0.348      | 0.267    |
| conv2 | 0.320   | **0.334**     | 0.303      | 0.221    |
| conv3 | 0.429   | **0.443**     | 0.407      | 0.326    |
| conv4 | 0.385   | **0.401**     | 0.369      | 0.316    |
| conv5 | 0.292   | **0.318**     | 0.282      | 0.203    |
| FC6   | 0.344   | **0.377**     | 0.325      | 0.235    |
| FC7   | 0.534   | **0.579**     | 0.541      | 0.434    |
| FC8   | 0.579   | **0.610**     | 0.543      | 0.446    |

The learned receptive field layer adapts receptive field sizes by spatial location, with larger fields in the periphery and smaller fields centrally (Figure 7). Despite some receptive fields being smaller than one pixel, all electrodes still contribute information, with single pixels used when needed. This effect is likely due to the model's limited $96 \times 96$ field of view, constraining pixel allocation for very small fields.

We also ran model ablations to assess the effect of different loss functions (discriminator, pixel, and VGG loss) on performance (Figure 11). Additionally, we trained models on region-specific data (V1, V4, IT) to explore how brain region training affects reconstruction (Figure 12). These experiments highlight the importance of individual brain regions and the adaptability of models trained on occluded data.

### 4.1.2 Spatial occlusion analysis

Spatial occlusion analysis was conducted to identify the importance of different brain regions (V1, V4, IT) in the reconstruction process. This analysis involved occluding specific spatial regions of the neural response inputs and examining the effect on the quality of the reconstructed images.

| Stimuli | V1+V4+IT | V1 | V4 | IT |
|---------|----------|----|----|----|

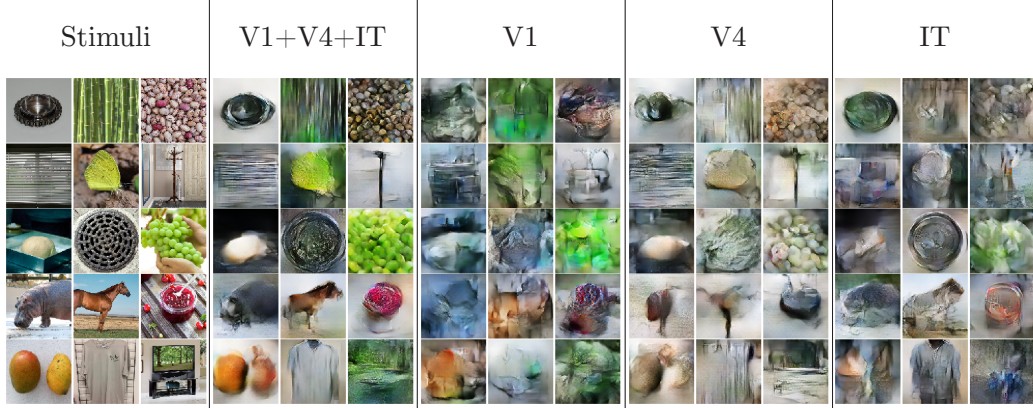

Figure 2: Spatial occlusion analysis of spatial model as explained in Section 4.1.2. Title above column means included brain region.

Figure 2 provides visualizations of example stimuli and their corresponding reconstructions when inputs from different brain regions (V1, V4, IT) were selectively occluded. The columns represent the regions of interest, while the rows present different example stimuli. For each region, the neural responses were set to their baseline values (pre-stimulus onset) for the occlusion procedure.

To quantify the impact of occlusion on reconstruction quality, feature correlations with the original images were calculated using the pre-trained AlexNet, as shown in Figure 8. The bar plot depicts feature correlations across different AlexNet layers for reconstructions derived from V1, V4, IT, and all regions combined.

## 4.2 Spatiotemporal occlusion analysis

Spatiotemporal occlusion analysis evaluates how neural responses from different time windows contribute to reconstruction quality, offering insights into the brain's temporal processing of visual stimuli.

Figure 3 shows example stimuli and their reconstructions when multiple time windows are occluded, with only one time window (highlighted in yellow) being included in the model during inference. Each column represents a different set of time windows being occluded, with the first column showing the reconstruction using all time windows.

The colorfulness of reconstructions from V1, V4, and IT was evaluated using the Hasler and Süsstrunk metric. Figure 4 shows the colorfulness scores for each brain region and combined data. V1-constrained reconstructions had the highest colorfulness scores and correlated strongly with early AlexNet layers (`conv1`, `conv2`), reflecting V1's role in processing basic features like edges and color. IT reconstructions, however, showed higher correlations with deeper AlexNet layers (`fc7`, `fc8`), which capture more abstract visual features.

These results highlight the hierarchical nature of visual processing, with V1 specializing in basic visual attributes and IT handling more complex features.

## 5 Conclusion

We presented a comprehensive approach to decoding naturalistic visual stimuli from neural responses using a fully convolutional neural network trained from scratch. The use of the THINGS dataset enriched our model's feature set, crucial for accurately interpreting brain representations. Our homeomorphic decoder, enhanced with an end-to-end inverse retinotopic mapping layer, effectively integrates spatial and temporal information, leading to high-fidelity and interpretable reconstructions. Our evaluations highlighted the spatiotemporal decoding model's superior performance, evidenced by high feature correlations with deep layers of pre-trained AlexNet. Spatial and spatiotemporal

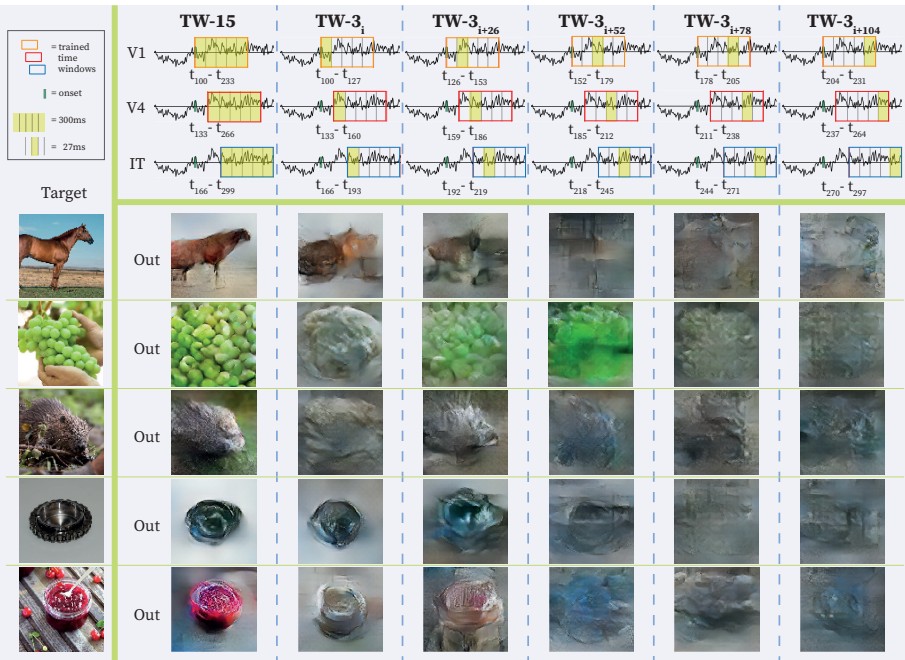

Figure 3: Spatiotemporal occlusion analysis. Yellow indicates the active time window, with others occluded.

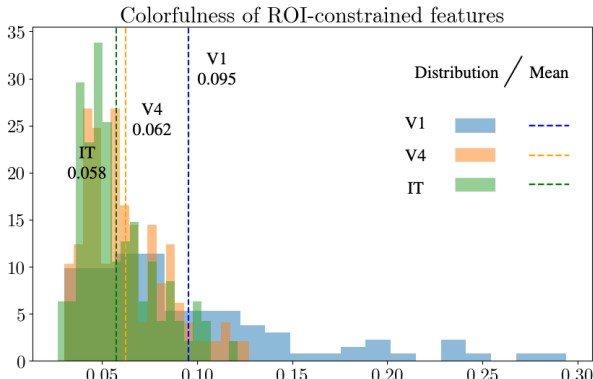

Figure 4: Distribution of colorfulness metrics across V1, V4, and IT-constrained reconstructions, calculated using the Composite Colorfulness Score (CCS) based on RGB channel differences.

occlusion analyses provided insights into the contributions of different brain regions and time windows, affirming the hierarchical nature of visual processing. The end-to-end inverse retinotopic mapping facilitated accurate estimation of receptive fields, aligning with neurophysiological findings and enhancing model transparency. This work advances neural decoding by offering a scalable and interpretable framework for reconstructing high-quality images from brain signals. Future research will explore more sophisticated architectures, further integrating temporal information and applying this framework to other sensory modalities.

**Broader impact**

Neural decoding models for reconstructing naturalistic images deepen our understanding of the link between neural activity and perception, with promising applications in visual neuroprosthetics. This study specifically reuses data originally collected as part of an initiative aimed at restoring sight, adhering to ethical best practices by maximizing insights from a single dataset and minimizing

the need for additional animal studies. This approach aligns with responsible research standards, balancing innovation with animal welfare.

However, caution is needed when applying these models to human neuroprosthetics due to the complexity of human brain activity and behavior. Human behavior encompasses interactions with the environment, complex motor actions, and neuroplasticity, elements that models trained exclusively on data from static image viewing may not fully capture. For instance, visual feedback from motor activities adds layers of complexity beyond current model capabilities. Additionally, using these models to identify stimulation sites for neuroprosthetics may not replicate natural neural responses precisely, highlighting the need for experiments/clinical trials with human subjects, nuanced model development and careful interpretation of outputs.

**Limitations**

This study applies invasive MUA recordings in macaques, expanding on prior work with fMRI by offering higher signal quality and detail. However, the applicability of these results to non-invasive techniques like fMRI remains limited due to their lower signal-to-noise ratio and less detailed recording capabilities compared to electrode arrays. This distinction is important, as invasive neuroimaging remains rare in human research.

Additionally, transitioning this framework to human intracranial applications poses challenges, including potential scar tissue formation, immune responses, and device rejection over long-term recordings. Anatomical differences, such as vascular structures, may further impact device placement and stability. Future work could explore these adaptations for broader applications, including brain-computer interfaces (BCI) and neuroprosthetics for individuals with acquired blindness, for which, careful regulatory guidance and additional research will be essential.

**Acknowledgments**

This work was supported by three grants of the Dutch Organization for Scientific Research (NWO): STW grant number P15-42 'NESTOR', ALW grant number 823-02-010 and Cross-over grant number 17619 'INTENSE' and grant number 024.005.022 'DBI2', a Gravitation program of the Dutch Ministry of Science, Education and Culture; the European Union's Horizon 2020 research and innovation programme: grant number 899287, 'NeuraViper'; the Human Brain Project, grant number 650003.

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

# A   Appendix / Supplemental material

## A.1   U-net architecture

**Contracting path**: The contracting path follows the typical architecture of a convolutional network. It consists of repeated application of two $3 \times 3$ convolutions (unpadded convolutions) followed by a ReLU activation and a $2 \times 2$ max pooling operation with stride 2 for downsampling. At each downsampling step, the number of feature channels is doubled.

**Expansive path**: Every step in the expansive path consists of an upsampling of the feature map followed by a $2 \times 2$ convolution ("up-convolution") that halves the number of feature channels, a concatenation with the correspondingly cropped feature map from the contracting path, and two $3 \times 3$ convolutions, each followed by a ReLU activation. The cropping is necessary due to the loss of border pixels during convolutions in the contracting path.

**Skip connections**: Skip connections are introduced between the contracting and expansive paths to combine low-level features with high-level features, facilitating finer reconstruction of the output image.

The input to the U-net is the retinal embedding $E$ of dimensions $15 \times 96 \times 96$ (15 channels and $96 \times 96$ spatial resolution) and the output is the reconstructed visual stimulus $S$ of dimensions $\text{RGB} \times 96 \times 96$. The U-net architecture enables detailed reconstruction by preserving spatial information through its symmetrical structure.

Table 2: U-NET Layers

| Layer | Shape | Configurations |
|---|---|---|
| Input | $96 \times 96 \times 15$ | - |
| Conv2d 1 | $48 \times 48 \times 64$ | kernel_size=4, stride=2, padding=1 |
| LeakyReLU 1 | $48 \times 48 \times 64$ | negative_slope=0.2, inplace=True |
| Identity | $48 \times 48 \times 512$ | Identity parameters: count=512, depth=512 |
| Skip 1 (Conv2d $\rightarrow$ ConvTranspose2d) | $48 \times 48 \times 512$ | See detailed breakdown |
| Skip 2 (Conv2d $\rightarrow$ ConvTranspose2d) | $48 \times 48 \times 256$ | See detailed breakdown |
| Skip 3 (Conv2d $\rightarrow$ ConvTranspose2d) | $48 \times 48 \times 128$ | See detailed breakdown |
| Skip 4 (Conv2d $\rightarrow$ ConvTranspose2d) | $48 \times 48 \times 64$ | See detailed breakdown |
| ConvTranspose2d | $96 \times 96 \times 3$ | kernel_size=4, stride=2, padding=1 |
| Sigmoid | $96 \times 96 \times 3$ | - |
| Output | $96 \times 96 \times 3$ | - |

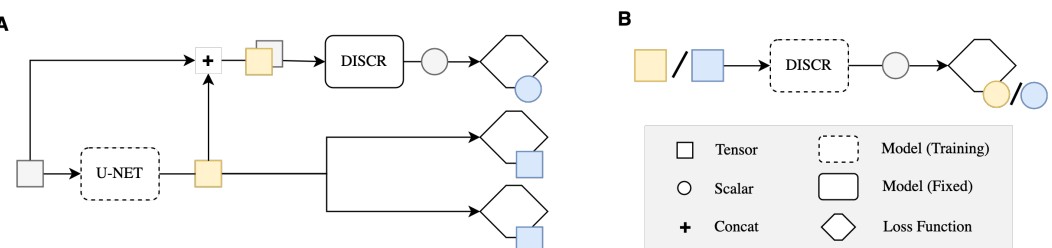

Figure 5: Overview of how the main reconstruction model is trained. **A.** The U-NET component is trained with a stack of 2D tensors (illustrated in grey) as input. These tensors are processed to produce reconstructions (depicted in yellow). The difference between the reconstructions and the target stimuli (represented in blue) are computed using the adversarial loss, feature loss, and pixel loss. **B.** Concurrently, the discriminator component undergoes its training phase. It evaluates the reconstructed outputs from the U-NET (labeled as 'fake images') alongside the original target images (labeled as 'real images'). This evaluation plays a critical role in calculating the Adversarial Loss, which is instrumental in guiding the parameter updates for the U-NET. This synergistic training approach ensures the progressive enhancement of the U-NET's ability to generate increasingly accurate and realistic reconstructed images.

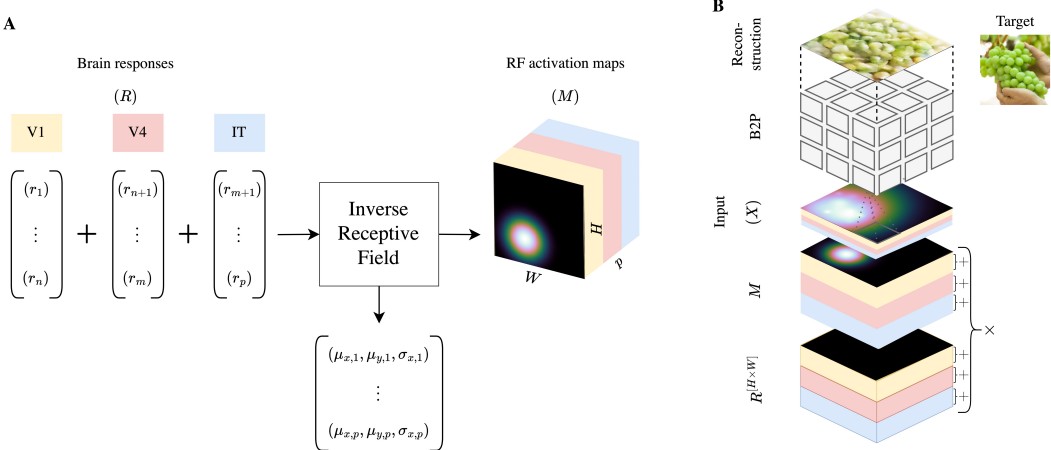

Figure 6: **A.** The inverse receptive field layer produces for each brain response $r \in \mathbb{R}$ an RF activation map ($M$) (also known as the embedding layer $E$) by using the learnable parameters ($\mu_x, \mu_y, \sigma$) in conjuction with the width ($W$) and height ($H$) of the desired model inputs ($X$) with a 2D Gaussian function. **B.** Let $R^{[H \times W]}$ be a matrix in $\mathbb{R}^{\mathbb{H} \times \mathbb{W}}$ such that each entry is $r$. $R^{[H \times W]}$ is multiplied element-wise with its corresponding $M$, and then stacked based on its electrode number, resulting in 15 $X$ in total (7 for V1, 4 for V4, and 4 for IT).

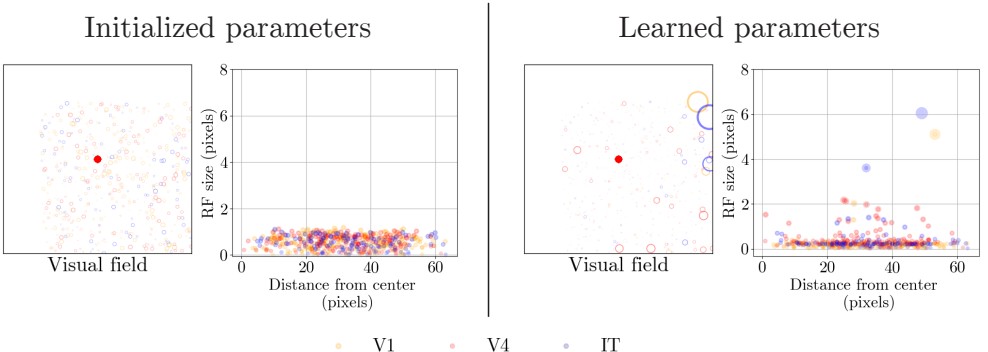

Figure 7: The learned 2D Gaussian parameters as spatial receptive field maps for mapping the neuronal signals in visual space as input for the reconstruction model. The "Visual field" shows the learned mappings in 2D space. The plot adjacent shows the variations in size of these RFs as a function of distance from the foveal center, highlighting how the learned RFs expands with increased eccentricity.

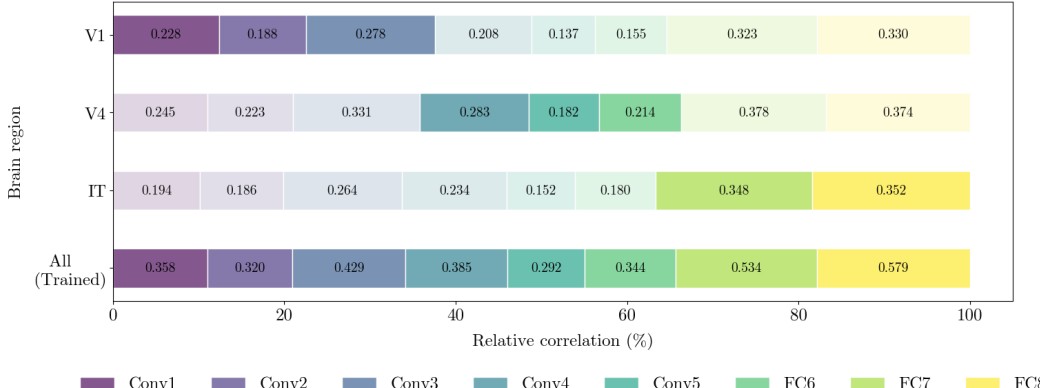

Figure 8: Relative correlation analysis of roi-constrained reconstructions with AlexNet features. This figure shows the relative correlation coefficients between features from roi-constrained reconstructions (V1, V4, IT) and corresponding AlexNet layers, normalized per brain region for fair comparison. Higher relative correlations are indicated by deeper colors and larger bars, marking the roi reconstruction with the closest match to each AlexNet layer's processing characteristics.

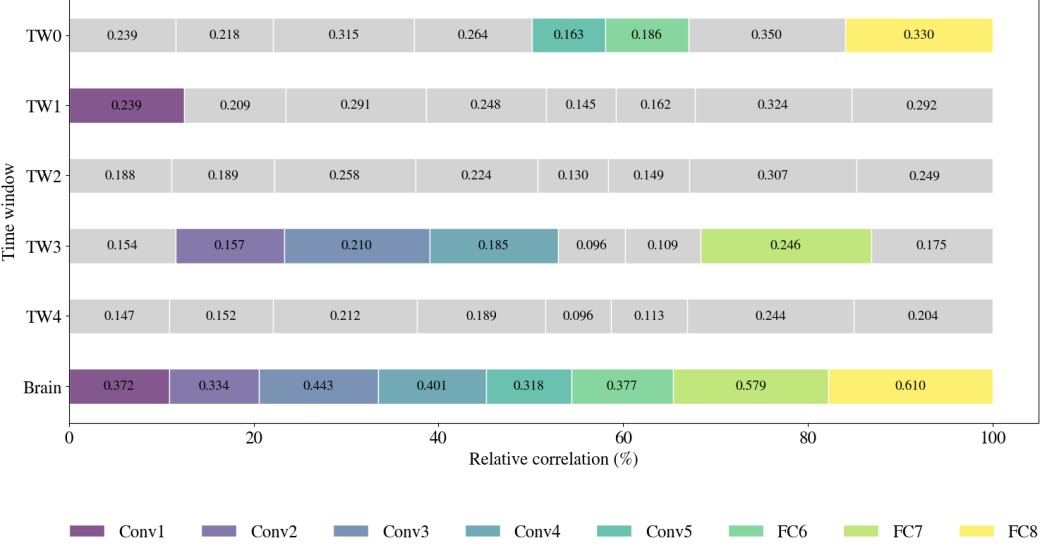

Figure 9: Temporal relative correlation analysis across AlexNet layers. This figure illustrates relative correlation coefficients across multiple time windows and AlexNet layers, with color and bar size representing the highest relative (not absolute) correlations per brain region. The x-axis is normalized, allowing direct comparison of relative contributions across time points.

AlexNet correlations of ROI and time-constrained reconstructions

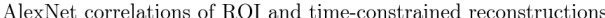

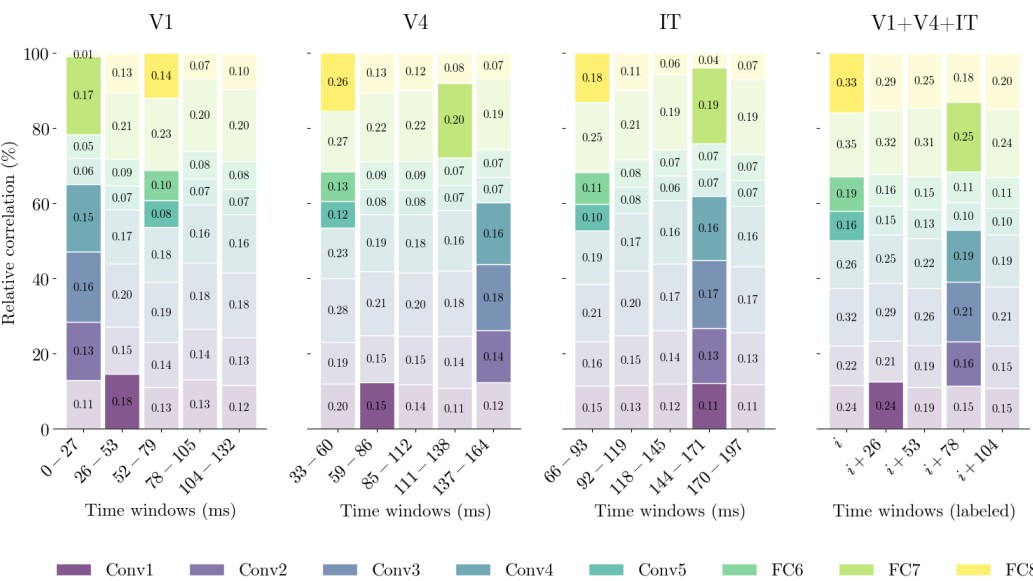

Figure 10: Correlation of features across time and brain, where $i$ is varying timepoints of the initial timewindow for each ROI (V1: 0-27ms, V4: 33-60ms, IT: 66-93ms) after stimulus onset and +26 is the 26 shift of all of the three windows.

Model ablations

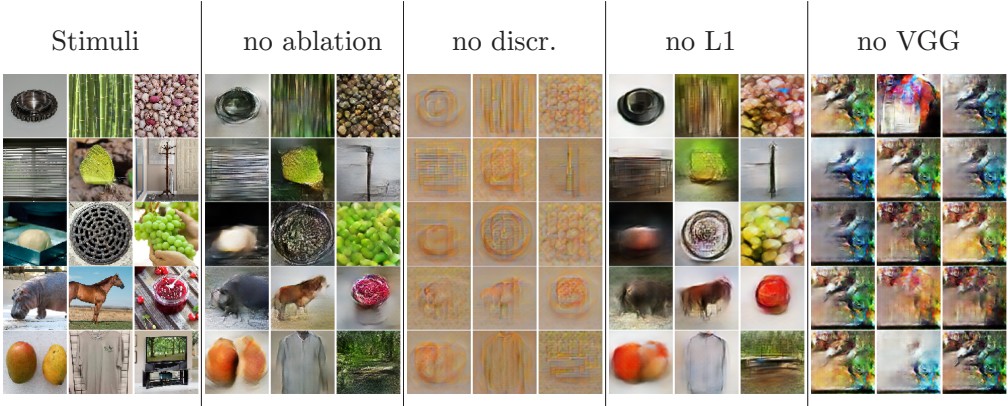

Figure 11: Training model with ablated components.

Model trained on brain region of interest

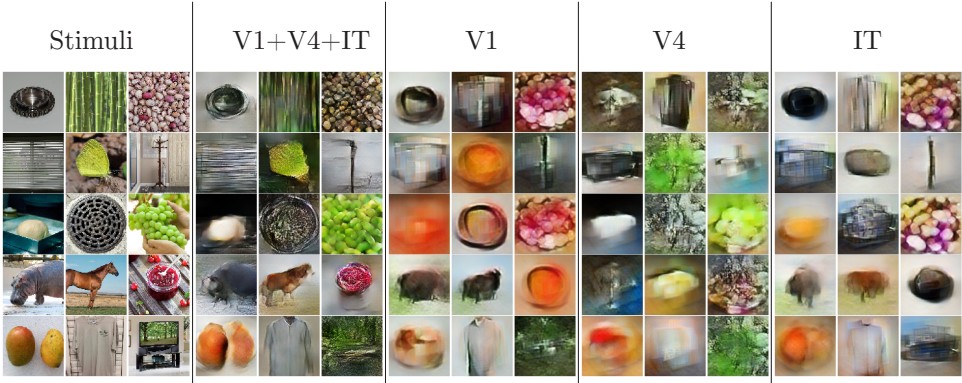

Figure 12: Training model on various brain regions.

