# OpenReview forum: "MonkeySee: Space-time-resolved reconstructions of natural images from macaque multi-unit activity"
_NeurIPS.cc/2024/Conference — NeurIPS 2024 poster_

### Official Review · Reviewer_gW2o · 2024-07-07

**Soundness:** 2
**Presentation:** 3
**Contribution:** 2
**Rating:** 6
**Confidence:** 4

**Summary:**

In this paper, authors record multi unit activity (MUA) from macaque ventral stream (V1, V4 and IT) and train a CNN decoder to reconstruct the visual stimuli. The authors present three variations decoding attempts: a baseline CNN decoder that maps MUA directly to image space, and two U-Net based decoders that differ in the fact that the first only uses spatial information (averaging over the time dimension), while the latter is time-resolved (spatiotemporal). Author find that the spatiotemporal decoder yield the highest performances as evaluated via feature correlation and present occlusion analysis (either masking brain regions or time windows).

**Strengths:**

The paper tackles an established field such as stimuli reconstruction using electrophysiological neural signals (MUA) as opposed to the more common fMRI techniques. This grants superior time-resolution which offers ways for novel experiments. Indeed author showed interesting effects of decoding from different time windows in different brain regions.
Furthermore the paper is in general well structured and presented.

**Weaknesses:**

- The paper present some inconsistencies between Text and Figures.
    * In line 267 it is said that the spatiotemporal model achieve the highest performance, however Table 1 reports the End-to-End models as the top scorer.
    * In line 288 author link to Figure 9 while the correct one should be Figure 3. However, description of Figure 3 is still inconsistent as in line 290 authors say that last column report full-data reconstruction while Figure 3 has it in the first column.
    * In Figure 8 meaning of x-axis is unclear. Moreover, Figure caption states that "Deeply colored points indicate higher correlations [...]", however it is unclear how this is assessed since for example the V4 row has higher correlation with fc7 and fc8 than IT (which is instead marked). Similar remarks hold for Figure 9, which has an incomplete caption (a trailing "The").
- Hyperparameters for decoder loss (sec. 3.6.2) are missing.
- Authors introduce the "Learned Receptive Field" (LRF) layer claiming it "enhances understanding of (model) structure and interpretive capacity", however no discussion about is presented. Only related result is Figure 7, which is never mentioned in the main text and presents somewhat puzzling results (see Questions section of this review).
- In the occlusion analyses, author perform model *inferences* on truncated data, however a model *trained* on the same occluded data might compensate the measured decrease in performances.
- Paper lacks comparisons with existing decoding techniques, albeit originally introduced for other - but structurally similar - modalities such as EEG signals (see Y. Bai et al., DreamDiffusion (2023)).

**Questions:**

- Figure 7 (not discussed in the main text) reports the learned 2D receptive field. Most learned RF seem to have a smaller-than-one pixel size, does this imply that no information from that electrode is used at all?
- Why do authors refer to their decoding technique as "homeomorphic"?
- Would a model trained on occluded data still performed as poorly as the results presented in the paper?
- Could the difference in measured performances between the baseline and homeomorphic model be simply attributed to the fact that the full model has higher complexity (more parameter), as the baseline decoder only leverages *half* of the U-Net architecture?

**Limitations:**

The authors have adequately addressed the limitations of their study.

---

> ### Author Rebuttal · Authors · 2024-08-07
>
> Thank you for your detailed review and constructive feedback. We appreciate the opportunity to clarify and address your concerns.
>
> **Addressing Mentioned Weaknesses:**
>
> **Text and Figures:**
>
> - **Line 267 and Table 1**: We apologize for the oversight. There was a mistake in the header of Table 1. The correct table should have the "End-to-End" and "Spatiotemporal" headers switched. The rest of the text is correct in stating that the spatiotemporal model achieved the highest performance overall, consistent with Figures 8 and 9. We will fix the table header.
>
> - **Figure references**: Line 288 should refer to Figure 3. We will correct the description to accurately reflect the columns. Thank you for pointing this out.
>
> - **Figures 8 and 9**: In Figure 8, the x-axis represents relative correlation, showing the percentage of correlation when all correlations add up to 1 per brain region. This ensures a fair comparison by normalizing values per brain region. The color coding represents the highest relative correlation, not the highest absolute correlation, also depicted by larger partial bars. We will clarify the x-axis meaning and provide a detailed explanation of the color coding and correlations. The caption for Figure 9 will be adjusted as well.
>
>
> - **Hyperparameters for decoder loss**: The hyperparameters for the training are mentioned in section 3.6.1: “We used the ADAM optimizer with a learning rate of 0.002 and beta coefficients of 0.5 and 0.999 to enhance convergence. The loss function included a discriminator loss (α_discr) at 0.01, VGG feature loss (β_vgg) at 0.9, and L1 pixel-wise loss (β_pix) at 0.09 to balance error sensitivity.”
>
>
> - **Learned Receptive Field (LRF) layer**: We thank the reviewer for this point. Figure 7 demonstrates that the LRF layer adapts during end-to-end reconstruction, showing smaller receptive fields in the center and larger ones in the periphery, indicating that information is used differently based on spatial location. This suggests that the LRF learns a mechanism similar to the brain while reconstructing end-to-end. We will discuss these results in the main text with more emphasis.
>
>
> - **Occlusion analyses**: Our goal with the occlusion analyses was to highlight the spatial and temporal features carried by neurons, not necessarily to compare reconstruction quality. We agree that training a model on occluded data could yield different results and will address this in the discussion. We did train the model on occluded data (see the second figure in the rebuttal).
>
>
> - **Comparisons with existing decoding techniques**: We currently compare our model against two baselines: the Brain2Pix model from Le et al. (2021) and an improved version of Shen et al. (2019). These models represent the state-of-the-art in the field. We will emphasize this in the manuscript and cite the mentioned paper (Y. Bai et al., DreamDiffusion, 2023) as well as any other relevant ones in the revised manuscript.
>
>
> **Answer to Questions:**
>
>
> - **Figure 7**: Many learned receptive fields (RFs) are indeed smaller than one pixel. However, this does not mean no information from that electrode is used. If an RF estimate is less than a pixel, the model uses a single pixel for that RF, still containing necessary information. This results from the limited field of view of the model (96 by 96 pixels) being unable to accurately assign sufficient (integer number of) pixels to electrodes with relatively smaller receptive fields (e.g., those with visual angles less than one degree of arc).
>
>
> - **Homeomorphic decoding**: We referred to our technique as "homeomorphic" due to its topography-preserving nature, formulating the neuron-to-pixel mapping problem as an image-to-image translation problem after a (learnable) retinotopic projection. The model preserves the inherent topography of data throughout all the network layers, focusing on processing local features.
>
>
> - **Occluded data**: While training the model on occluded data might yield different results, our objective was to demonstrate feature dependencies of a model trained on full brain data, not to evaluate absolute model performance on occluded data. We also trained separate reconstruction models on V1, V4, and IT regions, in addition to the combined model, which will be included in the manuscript.
>
>
> - **Model complexity**: The difference in performances between the baseline and homeomorphic model can be attributed to the higher complexity of the full model. Our baseline model, based on Shen et al. (2019), uses only half of the U-NET architecture, while our enhanced generator includes a full U-NET with more parameters. While increasing the baseline's complexity might deviate from strict comparisons with existing models, it illustrates how increased model complexity affects results. We will include an additional run with a full U-NET in the revised manuscript, which showed improved performance compared to Shen et al. but still fell short of the full homeomorphic model’s performance.
>
>
> Thank you again for your valuable feedback. We will revise the manuscript to address these points, improve clarity, and include the new analyses in the manuscript and the rebuttal images.
>
>
> **References:**
>
>
> - Guohua Shen, Kshitij Dwivedi, Kei Majima, Tomoyasu Horikawa, and Yukiyasu Kamitani. End-to-end deep image reconstruction from human brain activity. *Frontiers in Computational Neuroscience*, 13:432276, 2019.
> - Le, L., et al. (2022). Brain2pix: Video frame reconstruction from brain activity. *Frontiers in Neuroscience*, 16, 940972.
> - Y. Bai et al., DreamDiffusion (2023).

---

> > ### Comment · Reviewer_gW2o · 2024-08-12
> >
> > I thank the authors for providing additional material in their rebuttal in such short time and for carefully addressing my specific concerns with a detailed reply. I believe their work was strengthened by the new material and most importantly the new result that even a complexity-matched model "still fell short of the full homeomorphic model's performance". I will increase my review score to reflect this change.

---

> ### Author Response · Authors · 2024-08-12
>
> Thank you very much. We greatly appreciate it.

---

### Official Review · Reviewer_R5Jx · 2024-07-09

**Soundness:** 3
**Presentation:** 2
**Contribution:** 3
**Rating:** 6
**Confidence:** 4

**Summary:**

SETTING: Decoding (reconstructing) observed images via Utah-array recordings made from the visual cortex of a macaque.

APPROACH: A GAN, but taking (transformed--see below) neural data as input, and with some additional losses.  In particular, in addition to the standard adversarial loss, the generator/decoder is aligned with VGG-19 at various layers, and its output is penalized with a reconstruction penalty (mean absolute error).

Raw neural data are not passed into the generator/decoder (except in the baseline model); instead, multi-unit activity is first mapped to a retinotopic/pixel space, either with (learned) isotropic Gaussian receptive fields, or by mapping to features of a pre-trained Inception network.

RESULTS: The data and model evidently support high-quality reconstructions (Fig. 1).  The authors believe the decoder with a spatiotemporal inverse retinotopic map works best.  They also perform qualitative experiments to determine which cortical areas (V1, V4, IT) and time windows contribute what to image decoding.

**Strengths:**

The image reconstruction is, to my knowledge, state of the art.  The authors also convincingly demonstrate the importance of the retinotopic mapping.  Finally, the experimental setup (15 Utah arrays in one animal!) is heroic.

**Weaknesses:**

I put much of the details of these weaknesses into the questions (below).

It's not clear what we learn from the results following Figure 1.  Part (but not all--see the questions) of the problem may be due to the presentation.  The model itself is also not presented clearly: The model figures (Figs. 5 and 6 in the Appendix--not referenced in the main text) do not cover the first type of inverse retinotopic map, and introduce yet more symbols not appearing in the text.  The equations in the text appear incomplete (see questions).

**Questions:**

> The number of arrays is certainly much larger than what is used by almost any research group.  It would be very helpful to see how results scale with number of neurons (really, electrodes).  Such a figure would also perhaps help make sense of Fig. 2: how much of the degradation in image reconstruction there is due to using fewer neurons, and how much due to using neurons only from this or that area?  (If there isn't time to run this experiment, perhaps the authors have some other way of answering this question.)

> Along these lines: The authors somehow only get ~600 neurons from these fifteen arrays, apparently because they reject neurons with "intraclass correlations" less than 0.4.  What are these ICCs?  (What are the classes?)  Does this really improve performance?  How did the authors come up with the threshold of 0.4?

> Generically, I understand why one would want to map from electrode space into retinotopic/pixel space (basically so that the model can be a UNet from there on out.)  But I don't understand the explanation of the "inverse retinotopic mapping" the authors given is Section 3.  The authors give formulae for computing the retinal embedding E and say that the parameters of these functions are learned.  But where is E subsequently used?  For the "pre-trained mapping," a cost function is given, but E doesn't appear in it.  Instead, it contains yet more weights, in this case mapping from a pretrained Inception network to neural responses.  (In section 3.3.1, the letter W or w is used with seven different subscripts, some of them seemingly referring to the same weights and others not.)

In the appendix, there is a figure (not referenced in the main text!) that seemingly corresponds to the end-to-end inverse retinotopic mapping.  But E doesn't appear in this figure either.  The caption has a broken reference to an equation.

Can the authors clarify this part of the model?

> The authors state in the text that the spatiotemporal model achieves the highest correlations with AlexNet features, but Table 1 shows it to achieve the lowest (excluding the baseline; the end-to-end model achieves the highest correlation).  Is the text or the table in error?

They also claim that the spatiotemporal model qualitatively outperforms the other models (Fig. 1), but that claim is not obvious to this reader's eye.  (The horse and the butterfly, e.g., look worse; may others are a push.)

> Can the authors confirm that the 100 images used for testing were never used during training?

> What conclusions do the authors draw from Fig. 3?

In any case, the results would seem to depend strongly on the values assumed for synaptic delays:  This determines what time window in IT corresponds to what time window in V1, etc. (otherwise the windowing scheme is essentially random).  Can the authors justify high confidence in this synaptic-delay time?

MINOR:
> Section 4.1.2 examines the effect on decoding of setting certain areas of visual cortex to their baseline values.  Why is this called "spatial occlusion"?

Also, the column headings are described as identifying which brain region has been excluded, rather than included, but that obviously can't apply to the second column ("V1 + V4 + IT").  I think the text is wrong here.  (But if not, perhaps the authors could reverse the labels, letting all column headings state which areas are *in*cluded.)

> Figs. 8 and 9: I don't understand the lightness/darkness color scheme, which doesn't seem to correspond to the correlation values listed in the figure.  Can the authors clarify this?

> Why is the decoder called "homeomorphic"?

> "...spatially downsampling data to 8 Hz..."
What does this mean (and how could spatial sampling be denominated in Hertz?)?

> "...x_e and y_e denote the spatial coordinates of electrode e...."  I think it would be less confusing to say the "retinotopic coordinates corresponding to electrode e," or something similar.

> The reference to Fig. 9 (l. 288) should be to Fig. 3.

> The supplementary figures should be referenced in the main text.  (Figure 6, e.g., would be very useful to see while reading Section 3.4.)

> Fig. 3: "...with the final column representing the full data reconstruction" -> "...with the *first* column....."

**Limitations:**

N/A.

---

> ### Author Rebuttal · Authors · 2024-08-07
>
> Thank you for your thorough and thoughtful review. We appreciate your positive feedback on our contributions and the detailed suggestions for improvement. Your insights on our decoding approach are invaluable. We are glad to hear that our retinotopic mapping approach is recognized for its importance. We also appreciate your acknowledgment of our experimental setup. Below, we address your concerns and clarify the points raised.
>
> ### Model Presentation:
> We have fixed the figures to match the text and will clarify the presentation of our model and ensure that all relevant figures (Figs. 5 and 6) and equations are clearly referenced and explained in the main text. The inverse retinotopic mapping and its role in the model will be more clearly described.
>
> ### Answer to Questions:
>
> **Scaling Results with Number of Neurons:**
> We agree that analyzing how results scale with the number of neurons would provide further valuable insights. Therefore, we provide new comparative results of reconstructing the images only using data from electrodes that are located in specific visual ROIs. The results of this new experiment can be found in the Rebuttal Figures Document. Specifically, we show the reconstruction results from the regions V1, V4, and IT, separately.
>
> **Threshold Explanation:**
> We apologize for the incorrect use of "Intraclass Correlation Coefficient" in our manuscript. The measure we used was a reliability score based on self-correlation, which evaluates the consistency of neural responses across repetitions. The threshold of 0.4 was selected to balance retaining sufficient electrodes while removing unstable signals. Approximately 600 electrodes remained from the original 1024, partly due to this threshold and partly because one electrode was broken.
>
> **What is E:**
> The inverse retinotopic mapping is learned by the model in an end-to-end manner. The retinal embeddings (E) are not pre-calculated in this end-to-end model, whereas in the spatial and spatiotemporal model they are. We apologize for the confusion as we have used E interchangeably with RFSimages in the figure and agree that it is confusing. We will fix this and clarify it in the manuscript.
>
> **Appendix Figure:**
> This figure is a depiction of how the inverse receptive field layer in the end-to-end model computes the retinal embeddings (E) based on input brain responses. Apologies for the broken reference, we will fix this in the manuscript.
>
> We have removed this claim of qualitative outperformance.
>
> **Test Images:**
> Yes, definitely. We confirm that the 100 test images were never used during training.
>
> **Figure 3:**
> Figure 3 illustrates that the models are capable of detecting temporal variations in the information carried by neurons. This finding is promising and warrants further investigation in future research. By occluding the time windows that are not highlighted in Figure 3, we infer that the neurons at the highlighted time points carry the specific visual information indicated by our model. As time progresses, we observe changes in this information. We will clarify this in the text in the manuscript.
>
> **Synaptic Delay:**
> We base this assumption of synaptic delay on literature (Kravitz et al., 2013).
>
> **Text and Table Consistency:**
> We will correct the inconsistency between the text and Table 1 regarding the spatiotemporal model’s performance and ensure the qualitative claims are supported by clearer visual evidence.
>
> **Training and Testing Data:**
> We confirm that the 100 images used for testing were never used during training and will explicitly state this in the revised manuscript.
>
> ### Minor Issues:
> We have addressed and fixed the minor issues mentioned in the paper:
> - The term "spatial occlusion" was used to indicate the model's inference when specific locations of the input are set to baseline values (i.e., no signal or activity). The occlusion occurs at the regions that we are not interested in. Baseline values are used instead of zero to account for existing noise.
> - The column headings in Section 4.1.2 are currently explaining which areas are included indeed. Will add it to the figure description.
> - In Figures 8 and 9, the x-axis represents relative correlation, showing the percentage of correlation when all correlations add up to 1 per brain region. This ensures a fair comparison by normalizing values per brain region. The color coding represents the highest relative correlation, not the highest absolute correlation, also depicted by larger bars. We will clarify the x-axis meaning and provide a detailed explanation of the color coding and correlations. The caption for Figure 9 will be adjusted as well. We added this explanation in the main text as well.
> - We referred to our technique as "homeomorphic" due to its topography-preserving nature, formulating the neuron-to-pixel mapping problem as an image-to-image translation problem after a (learnable) retinotopic projection. The model preserves the inherent topography of data throughout all the network layers, focusing on processing local features.
>
>   *Two shapes are said to be homeomorphic if there exists a continuous, one-to-one mapping between them that preserves the spatial relationships of points. If you can stretch or bend one shape into another without tearing or gluing parts together, the shapes are homeomorphic.*
>
> - Changed the reference from spatial downsampling in Hz to temporally downsampling data to 8 Hz.
> - Revised the description of electrode coordinates.
> - Fixed the reference to Fig. 3 and the labeling issues in Fig. 3.
> - Ensured all supplementary figures are properly referenced in the main text.
>
> Thank you again for your constructive feedback.
>
> - Kravitz, D. J., Saleem, K. S., Baker, C. I., Ungerleider, L. G., & Mishkin, M. (2013). The ventral visual pathway: An expanded neural framework for the processing of object quality. *Trends in Cognitive Sciences, 17*(1), 26-49.

---

> > ### Comment · Reviewer_R5Jx · 2024-08-12
> >
> > The authors have answered some of my questions but not all.  The inverse retinotopic mapping still isn't adequately explained (the authors still haven't said how, e.g., E_axy enters the objective function).
> >
> > It's still not clear what conclusions the reader is supposed to draw from Fig. 3.  The authors reply that "we infer that the neurons at the highlighted time points carry the specific visual information indicated by our model," but what information is this?  Can it be described even qualitatively?  E.g., "excluding TW-3 tends to make the images..."--what?  Blurrier?  If this analysis were redone with a different choice for synaptic delay (say, a factor of 2 larger), would we draw different conclusions?
> >
> > Still, the paper's strengths remain, and I stand by my original ratings.

---

> ### Author Response · Authors · 2024-08-14
> **We thank the reviewer for these clarification questions!**
>
> **Q1**. Let's first define the following:
>
> - S := visual stimulus
>   - S_real := ground-truth stimulus
>   - S_fake := reconstructed stimulus
> - R := neural response
> - E := retinal embedding
>
> After (pre-/end-to-end trained) inverse retinotopic mapping projects R onto E as in the (pre-/end-to-end trained) inverse retinotopic mapping subsection, pixel-to-pixel mapping reconstructs S_fake from E as S_fake = U_Net(E). Note that the U-Net architecture is provided in the appendix. Every loss component is a function of S_real and/or S_fake, which is how, e.g., E enters the objective function.
>
> We will make sure to thoroughly review and clarify the relevant subsections in the revised manuscript.
>
> **Q2**. Figure 3 demonstrates the temporal occlusion analysis results of five example stimulus-response pairs. While we can observe some qualitative trends in these results, we cannot easily draw general/strong conclusions from the figure alone.
>
> For example, it appears that the form/shape information is present mostly in the first time window. Likewise, color information seems to be mostly present in the second and third time windows.
>
> Also, quantitative results of the temporal occlusion analysis is provided in Figure 9 and 10, which complement Figure 3 and can be interpreted in a similar way to the spatial occlusion analysis results. That is, in terms of the internal representations of the AlexNet layer that show the highest correlation at a time window  (instead of only ROI as in the spatial occlusion analysis).
>
> Likewise, we will make sure to thoroughly review and clarify the relevant  subsections, figure captions, as well as correcting the figure references  in the revised manuscript.

---

### Official Review · Reviewer_d65o · 2024-07-11

**Soundness:** 4
**Presentation:** 3
**Contribution:** 4
**Rating:** 7
**Confidence:** 4

**Summary:**

The authors present a CNN-based decoder that illuminates the distinct information encoded in V1, V4, and IT neuronal populations. The decoding results are remarkably good, and their accomplishment in decoding natural images from neuronal-level signals is unprecedented and the best so far.

**Strengths:**

Their novel space-time resolved decoding technique and the learned receptive field layers are new and interesting. Perhaps most impressive of all is their unprecedented data, recorded simultaneously from 15 Utah arrays from a single monkey. It can be considered a milestone in neuroscience research.

**Weaknesses:**

Analytical innovation is relatively minor. I am not sure what new insights about the visual cortex we learned. Nevertheless, proving that it can be done (with 4-7 Utah arrays per area), and doing it beautifully is still a remarkable feat that should be admired and applauded.

**Questions:**

No specific questions.

**Limitations:**

No concerns.

---

> ### Author Rebuttal · Authors · 2024-08-07
>
> Thank you for your thoughtful and positive review. We appreciate your recognition of our novel space-time resolved decoding technique and the data collected from 15 Utah arrays. Regarding the mentioned weakness, our primary aim was indeed to demonstrate the feasibility and beauty of using these advanced models for studying the visual cortex. While the new neuroscientific insights reported in this work may seem limited, we believe this research is an essential step towards deeper understanding in the future. Your acknowledgment of our effort is greatly appreciated.

---

> > ### Comment · Reviewer_d65o · 2024-08-13
> > **Responses to Rebuttal**
> >
> > Thanks for your responses. Congratulation on your accomplishment.

---

### Official Review · Reviewer_skQc · 2024-07-15

**Soundness:** 3
**Presentation:** 2
**Contribution:** 2
**Rating:** 4
**Confidence:** 4

**Summary:**

This paper proposes a CNN-based decoder to reconstruct naturalistic images from macaque brain signals. To this end, the paper presents the Learned Receptive Field (LRF) layer to enhance the reconstruction and understanding of the model's structure and interpretive capacity. Here, the work aims to interpret brain activity by transforming low-level to high-level brain signals. It focuses on the readout characteristics of neuronal populations in areas V1, V4, and IT. The evaluations of the model are performed using the THINGS dataset, and the results demonstrate the effectiveness of an end-to-end model and provide interpretations, such as the crucial role of V1 in color processing.

**Strengths:**

- The integration of a U-Net architecture for the pixel-to-pixel mapping of retinal embeddings to visual stimuli is intriguing.

- The proposed method is end-to-end with different parameters, making it adaptable to the target problem.

- Though the method of the paper is data-specific, the proposed model’s results are somehow interpretable

**Weaknesses:**

- The proposed model is data-specific, which could potentially limit its application.

- The overall loss consists of multiple criteria, each with its weight, which might make the model harder to train. Specifically, mixing - adversarial loss with pixel and feature loss can make it challenging to control the training dynamics.

- The paper does not include a complete ablation analysis of the different modules.

**Questions:**

- Could you elaborate more on the motivation behind using Inception v1 for different cortical areas? How does this layer contribute to the generalization of the image reconstruction?

- Is the model generalizable to other types of neural data from different species or different types of recording equipment?

- In equation line 151, how does the standard deviation influence the spatial spread of the receptive field and overall reconstruction?

- In Pre-trained Inverse Retinotopic Mapping, why do we need two embeddings? Can’t the model handle it with a unified spatiotemporal module instead of having separate embeddings?

- What about employing an existing method to build a baseline, such as using a recurrent CNN or Vision Transformer? For a fair comparison, the evaluation should include some of the other state-of-the-art architectures.

- Is the quantitative comparison presented in Table 1 a standard manner of measuring performance? Is it used in the literature? If not, what would be other metrics, such as the R² score?

- What about the ablation analysis of the loss functions? Also, could you present the dynamic criterion of each loss during the training and validation phases?

- How did you set the hyperparameters of the model? Also, the full implementation details of the model are missing.

**Limitations:**

- I think the main limitation of the paper is the limited area of applicability of the proposed framework.

- The evaluation, both in terms of architecture and criterion, is missing to determine the necessity and effectiveness of all the modules of the model.

---

> ### Author Rebuttal · Authors · 2024-08-07
>
> Thank you for your detailed feedback. We appreciate your positive comments and are pleased to hear that you recognized our model's adaptability. Below, we address your points in further detail:
>
>
> **Weaknesses:**
>
> 1. **Data-specific model**: While our model was trained on the THINGS dataset, which is extensive and diverse, it has demonstrated robust performance on limited data, which is crucial given the challenges of collecting invasive monkey data. Our model is also generalizable to various types of neural data and recording equipment, similar to earlier models successfully applied to human fMRI data (Le et al., 2022).
>
>
> 2. **Loss functions and training dynamics**: Although our model employs multiple loss functions, their combination enhances training stability and performance. We have conducted additional ablation studies, which we will include in the revised manuscript, demonstrating the necessity of each loss component for optimal model performance.
>
>
> 3. **Complete ablation analysis**: We acknowledge the importance of comprehensive ablation analysis. We have now performed additional ablation studies, examining the contributions of the discriminator, L1, and VGG losses, and will include these results in the revised manuscript (also see figure 1 provided in rebuttal).
>
>
> **Questions:**
>
>
> 1. **Motivation for using Inception v1**: Inception v1 was chosen for its capability to capture features with varying receptive field sizes and complexities, aligning with the hierarchical processing in the ventral visual stream. This alignment allows effective generalization across visual stimuli and provides accurate predictions and meaningful MEIs for neurons in V1, V4, and IT. Previous studies (e.g., Yamins et al., 2014; Khaligh-Razavi & Kriegeskorte, 2014; Güçlü & van Gerven, 2015) have demonstrated the success of task-optimized DNN models in modeling visual areas.
>
> 2. **Model generalization**: Our model is designed to be generalizable to different types of neural data and recording equipment, as demonstrated by its successful application to human fMRI data (Le et al., 2022).
>
> 3. **Standard deviation in receptive fields**: The standard deviation models the size of the receptive field, directly influencing the accuracy of stimulus reconstructions. Precise modeling of receptive field sizes enhances the quality of the reconstructed images.
>
> 4. **Pre-trained Inverse Retinotopic Mapping**: The use of two embeddings corresponds to our model’s variants for spatial and spatiotemporal reconstructions. Each variant requires specific embeddings to optimize performance in their respective domains.
>
> 5. **Baseline comparisons**: We currently compare our model against two baselines: the Brain2Pix model (Le et al., 2022) and an improved version of Shen et al. (2019). These models represent the state-of-the-art in the field. We will emphasize this in the manuscript.
>
> 6. **Performance metrics**: The metrics presented in Table 1 are standard in the context of image reconstruction and have been used in previous studies. We will clarify this in the revised manuscript.
>
> 7. **Ablation analysis of loss functions**: We have now conducted ablation analyses on the discriminator, L1, and VGG losses. The results, showing decreased performance when these components are removed, will be included in the revised manuscript. Additionally, we trained separate models for V1, V4, and IT, in addition to the combined model, and will present these findings.
>
> 8. **Hyperparameter settings and implementation details**: Our hyperparameters were set based on existing literature and detailed in section 3.6.4. We used the ADAM optimizer with specified parameters and provided full architecture details in the supplementary material. The training process has proven robust to hyperparameter variations, as confirmed by our pilot experiments. We have corrected the link to the full source code in our official comment to the AC, adhering to NeurIPS guidelines.
>
>
> **Limitations:**
>
>
> 1. **Limited area of applicability**: In the revised manuscript, we will discuss potential extensions and adaptations of our framework, including applications in BCI communication and neuroprosthetics for individuals with acquired blindness.
>
> 2. **Evaluation of architecture and criterion**: We have addressed this in our responses above, providing additional evaluation and justification for our model's architecture and criteria.
>
> Thank you again for your insightful comments. We believe these revisions will significantly strengthen our manuscript and address the concerns raised.
>
> **References:**
>
>
> - Yamins, D. L., et al. (2014). Performance-optimized hierarchical models predict neural responses in higher visual cortex. *PNAS*, 111(23), 8619-8624.
> - Khaligh-Razavi, S. M., & Kriegeskorte, N. (2014). Deep supervised models may explain IT cortical representation. *PLoS Computational Biology*, 10(11), e1003915.
> - Güçlü, U., & van Gerven, M. A. J. (2015). Deep neural networks reveal a gradient in the complexity of neural representations across the ventral stream. *Journal of Neuroscience*, 35(27), 10005-10014.
> - Le, L., et al. (2022). Brain2pix: Video frame reconstruction from brain activity. *Frontiers in Neuroscience*, 16, 940972.
> - Shen, G., et al. (2019). End-to-end deep image reconstruction from human brain activity. *Frontiers in Computational Neuroscience*, 13, 432276.

---

### Author Rebuttal · Authors · 2024-08-07

We have included additional results in the form of figures as requested. The ablation study on various losses is presented in the figure titled "Model Ablations," along with the corresponding model losses for each ablated model in the figure titled "Ablation Losses." Although these were run with fewer epochs than usual, we were already able to observe the effects of the ablations.

Additionally, we have trained our model on different brain regions in response to reviewer gW2o's request, depicted in the figure titled "Model Trained on Brain Region of Interest." The figures and detailed results provide insight into the model's performance under these conditions.

We hope these additions address your concerns and provide a clearer understanding of our model's robustness and versatility. The rest of the rebuttals are addressed individually to each reviewer.

Thank you for your valuable feedback and guidance.

---

### Decision · Program_Chairs · 2024-09-25

**Decision:**

Accept (poster)

**Comment:**

In this paper, the authors present a method for image decoding from spatiotemporal information from neural recordings. The authors provide good reconstructions of the image, employing the temporal dimension for further accuracy as well as more insight into the brain. The authors also provide insight into the role of different regions in the decoding process. Some concerns where raised by the reviewers, and the author responses and clarifications should be part of the final document.